# The Effects of Different Slurry Concentrations and Wire Speeds for Swinging and Non-Swinging Wire-Saw Machining

Yao-Yang Tsai [1], Yi-Chian Chen [2], Yunn-Shiuan Liao [1], Chia-Chin Hsieh [2], Chung-Chen Tsao [2,*] and Chun-Yao Hsu [2,*] 

[1] Department of Mechanical Engineering, National Taiwan University, Taipei 10617, Taiwan; yytsai@ntu.edu.tw (Y.-Y.T.); liaoys@ntu.edu.tw (Y.-S.L.)

[2] Department of Mechanical Engineering, Lunghwa University of Science and Technology, Taoyuan 33306, Taiwan; ycchen@mail.lhu.edu.tw (Y.-C.C.); 05370465@me.mcu.edu.tw (C.-C.H.)

[*] Correspondence: aetcc@mail.lhu.edu.tw (C.-C.T.); cyhsu@mail.lhu.edu.tw (C.-Y.H.); Fax: +886-2-2096865 (C.-C.T. & C.-Y.H.)

**Abstract:** Slurry concentration and wire speed affect the yield and machining quality of ceramics ($Al_2O_3$) that are produced using wire-saw machining (WSM). This study determines the effect of slurry concentration and wire speed on the material removal rate (MRR), the machined surface roughness (SR), the kerf width, the wire wear and the flatness for swinging and non-swinging WSM. The experiments show that swinging WSM results in a higher machining efficiency than non-swinging WSM. WSM with swinging also achieves a peak MRR at a medium slurry concentration (25 wt%) and a higher wire speed (5.6 m/s) using the cutting conditions for the experimental region. However, slurry concentration and wire speed have no significant effect on the machined SR, the kerf width, the wire wear or the flatness for WSM with swinging mode.

**Keywords:** slurry concentration; wire speed; wire-saw machining; material removal rate; swinging

## 1. Introduction

Precision ceramics are used in a wide range of various advanced applications in the semiconductor and optoelectronics industries. Precision ceramics feature superior thermal resistance, wear resistance and corrosion resistance, and are lightweight, which gives them an advantage over metal. The development of precision ceramics will increase performance for the most advanced industries in the future. However, precision ceramics have a lower resistance to fracture than metals, which often undergo plastic deformation during machining, so the machined surface is marked with fractures in the form of extended minute cracks. It is difficult to control both machining precision and machined surface roughness (SR).

Ceramic is a key component in the semiconductor and optoelectronics industries. The slicing machining process for ceramics must be low cost, involve a short machining time, feature minimum loss of kerf and produce a good quality machined surface. A wire-saw uses abrasive cutting for the machining of hard, brittle materials and is efficient for cutting ceramics. WSM uses stainless steel wires to mobilize abrasive slurry and cut hard, brittle materials. A stainless steel wire for WSM can have a single or multiple strands. WSM dissipates less heat during machining, so there are small disturbed layers on the workpiece surface. An extremely fine slot is cut (wire diameter of 0.25 mm or even smaller) so there is less waste of machined materials. Similarly to a polishing process, WSM achieves good surface roughness if the cutting conditions are controlled. However, wire-saw machines can be less structurally rigid than conventional cutting machines.

There have been many experimented studies of WSM using various process parameters [1–3]. Teomete [1] showed that the material removal mechanism is related to trans-granular failure of the grains and inter-granular fractures within grains affect the surface quality. Lee et al. [2] showed that the break-in characteristics of diamond wire are strongly correlated with the wear behavior of diamond wire for single WSM. Qiu et al. [3] showed that process parameters, such as the magnetic field intensity, the feed rate for the workpiece, the running speed for the wire, the viscosity of the cutting fluid and the mass fraction of the magnetic abrasive grains, affect the cutting performance of a free abrasive. Bao et al. [4] noted that localized porous features are generated on the surface of the anodic oxide layer if multi-crystalline silicon ingots are sliced into wafers using abrasive electrochemical multi-wire sawing because the original surface undergoes anodic oxidation.

The total volume that is removed and the slicing efficiency for a brazed wire saw that uses different slicing parameters (diameter of wire-saw, diamond grain and diamond concentration) are respectively 2.2- and 1.5-times better than that for an electroplated wire-saw [5]. Li et al. [6] proposed a constant wire wear loss model, which gives good wafer quality and consumes 24.8% less wire than a conventional method for specific slicing parameters (length of ingot, wire speed, table feed speed, wire forward length and radius of ingot). Liu et al. [7] showed that the feed rate and the wire speed have a different effect on the depth of cut and the cutting force for resin-bonded diamond wire-sawing. WSM gives an increased MRR and decreased wire wear. The machined surface roughness, the kerf width and the flatness are indicators of the quality of WSM. Yeh et al. [8] showed that diamond wear increases as the bonding layer thickness decreases because grains protrude less as the thickness of the electroplated nickel layer increases, especially if the cutting speed and feed rate are increased for wire-sawing of sapphire ingots.

Hsu et al. [9] used the grey-Taguchi method to optimize the qualities of WSM (MRR, surface roughness, steel wire wear, kerf width and flatness) for ceramics using ultrasonic-assisted vibration. Clark et al. [10] used a capacitance sensor to measure the bow in the wire and converted the bow to a vertical cutting force for swinging WSM. Huang et al. [11] noted that the contact length changes significantly in one half of a swing period for wire-sawing with swinging using a square or circular ingot. Chen and Gupta [12] showed that a swinging motion decreases the contact length and the equivalent chip thickness by 50% for wire-sawing of a mono-crystalline alumina oxide wafer. Xu et al. [13] determined that the amplitude of the swing of the ingot for multi-wire sawing of mono-crystalline silicon wafers has no significant effect on the contact wire length but has a significant effect on the ingot feed speed. It was shown that a swinging amplitude for the ingot of 5° gives the best quality machined surface for WSM with swinging. This study determines the effect of machining parameters (slurry concentration, wire speed, swinging frequency and swinging angle) on the machining characteristics (MRR, roughness of the machined surface, kerf width, wire wear and flatness) for WSM with swinging.

## 2. Experimental Apparatus and Methodology

Figure 1 shows the experimental setup for the single wire-sawing system that is used for this study. The inverter (F) and the position of the belt (D) for the pulley (B) were adjusted to control the wire speed. The adjustment screw (I) was adjusted to achieve the proper wire (G) tension. The working load was varied by adjusting the balance weight (C) on the arm (A). A pump (H) was used to stir and supply slurry. The workpiece (J) was fixed to a jig (K). The motor (O) drives the rack (M) through a universal joint (N). The rack moves back and forth and drives the gear (L), which rotates in the same direction (gear (L)) and shares the same central shaft as the jig (K).

Figure 2a shows that the workpiece (J) swings back and forth. Figure 2b shows the length of contact between the wire and the workpiece during WSM. This swinging machining model exerts a uniform cutting force on each active grain and allows efficient disposal of chips. The main experimental conditions are listed in Table 1, wherein the slurry concentration is shown as a percentage of the weight

of the slurry grain (weight of grain/weight of slurry as a whole). The MRR and wire wear (WW) for each SWSM is calculated using Equations (1) and (2), which are defined as follows:

$$MRR = \frac{W_I - W_W}{t} \tag{1}$$

$$WW = \frac{D_I - D_W}{t} \tag{2}$$

where $W_I$ is the original weight before SWSM and $W_W$ is the actual weight after SWSM, $D_I$ is the original diameter before SWSM, $D_W$ is the diameter after SWSM and t is the period for SWSM. A tool microscope (Nikon, MM-40, Tokyo, Japan) acts as an optical ruler to accurately measure the wire wear to an accuracy of 1 μm. A tension meter (IMADA, DPRSX-10TR, Toyohashi, Japan) was used to measure the wire tension and working load. The surface morphology was determined using field emission scanning electron microscopy (JEOL JSM-6300 SEM, Tokyo, Japan).

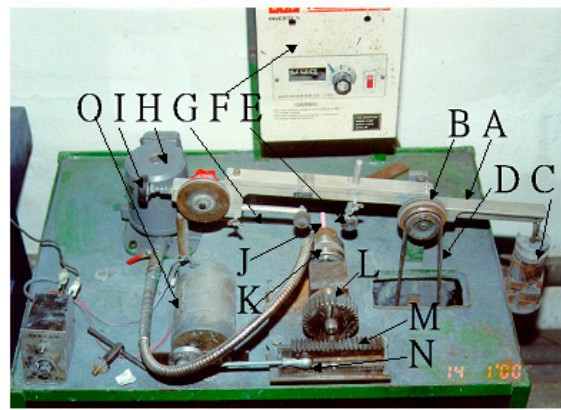

| Arm (A) | Pulleys (B) | Balance weight (C) |
|---|---|---|
| Belt (D) | Support roller (E) | Inverter (F) |
| Wire (G) | Pump (H) | Adjustment screw (I) |
| Workpiece (J) | Jig (K) | Gear (L) |
| Rack (M) | Universal joint (N) | Motor (O) |

**Figure 1.** Setup for the single wire-saw system.

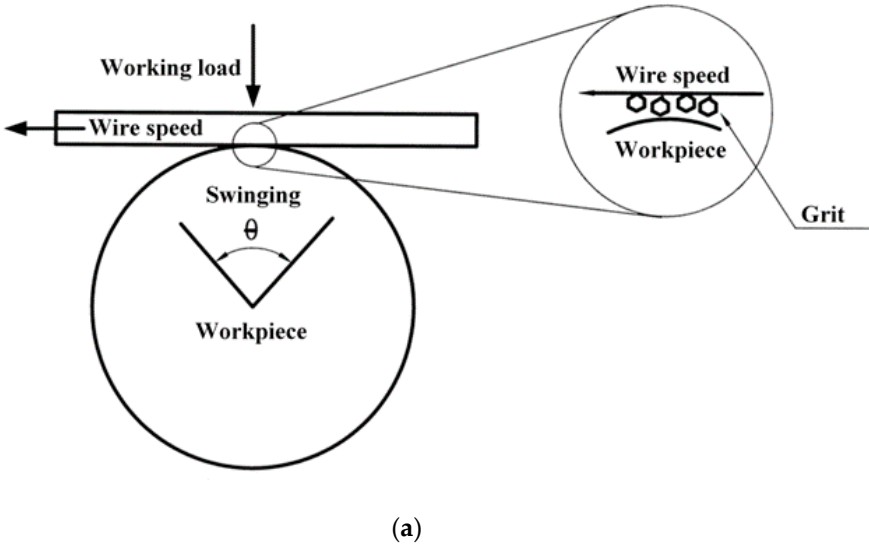

(**a**)

**Figure 2.** *Cont.*

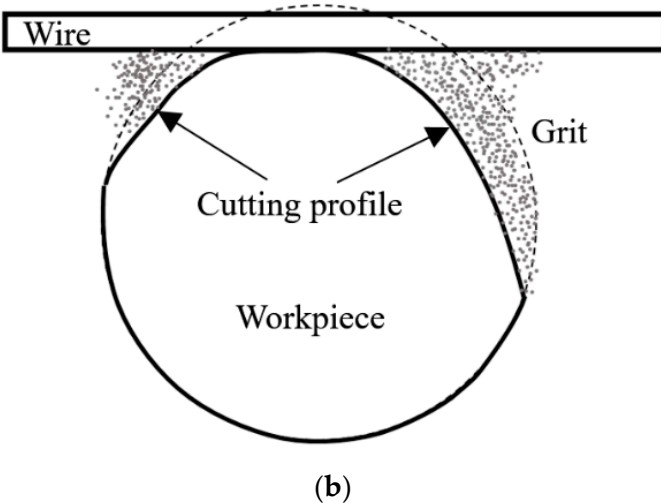

(**b**)

**Figure 2.** (**a**) Model of swinging wire-sawing and (**b**) contact between the wire and the workpiece.

**Table 1.** Experimental conditions.

| | |
|---|---|
| Workpiece | $Al_2O_3$ |
| Diameter (mm) | φ8 |
| Wire diameter (mm) | φ0.24 ± 0.05 (Stainless steel wire) |
| Slurry | Green silicon carbide (GC) + water |
| Grain size (mesh) | #600, #800 and #1000 |
| Wire tension (N) | 15 |
| Concentration (wt%) | 10, 14, 25, 35, 45, 56 and 66 |
| Wire speed (m/s) | 1.9, 2.8, 5.6, 6.4 and 7.8 |
| Working load (N) | 1.27, 1.76 and 2.35 |
| Swinging frequency (Hz) | 0.4, 0.8, 1.2 and 1.5 |
| Swinging angle (θ) | 40°, 60° and 90° |

## 3. Experimental Results and Data Analysis

### 3.1. Effect of Machining Parameters on MRR

The wire speed and slurry concentration are directly related to MRR for WSM. Figure 3 shows the experimental results for MRR for a non-swinging scenario for different wire speeds and slurry concentrations. The peak MRR is achieved at a higher slurry concentration and a higher wire speed. As the wire speed increases, the peak MRR is only achieved by increasing slurry concentration. This demonstrates that material is almost exclusively removed by the active grains in the slurry. The greater the slurry concentration, the greater is the number of active grains, so there is an obvious increase in the MRR. At a lower wire speed, active grains are removed more slowly from the machining zone so increasing the wire speed results in a greater MRR from the machining zone for a non-swinging scenario.

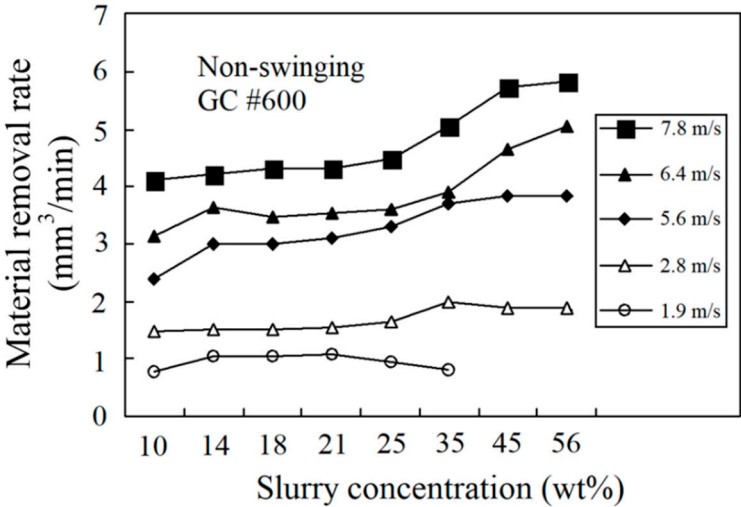

**Figure 3.** Effect of different wire speeds and slurry concentrations on material removal rate (MRR).

Figure 4 shows the relationship between swinging frequency, slurry concentration and MRR for swinging and non-swinging scenarios. The MRR is greatest at a swinging frequency of 1.2 cycle/s (Hz) because this swinging frequency allows active grains to enter and exit at a wire speed of 2.8 m/s and a slurry concentration of 25%. The contact between the wire and the workpiece for the swinging scenario is shorter than that for the non-swinging scenario, so each active grain produces a larger cutting force (or cutting pressure). The active grains also enter and exit the machining zone faster and there is more efficient disposal of chips for WSM. Therefore, WSM with swinging produces a higher MRR for a specific wire speed and slurry concentration.

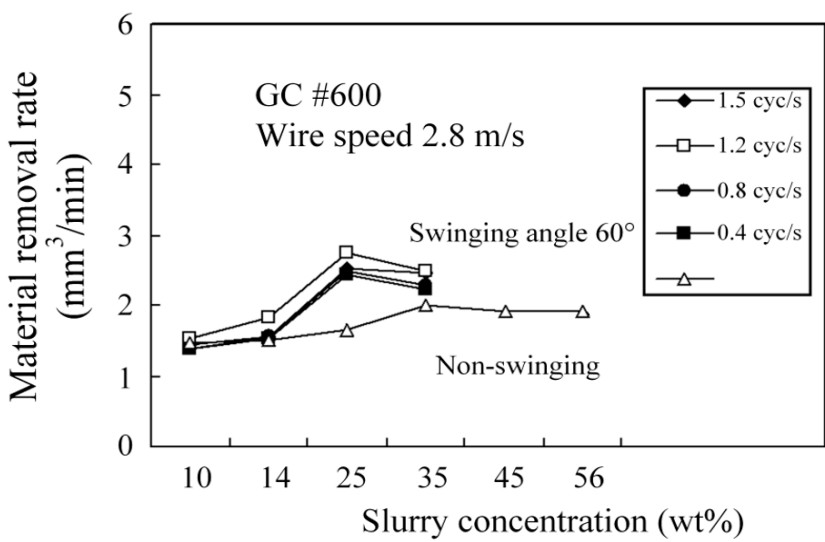

**Figure 4.** Comparison of swinging and non-swinging processes in terms of the MRR for different slurry concentrations and swinging frequencies.

Figure 5 shows the effect of the swinging angle and frequency on the MRR for swinging WSM. The results show that swinging angles of 40°, 60° and 90° have no significant effect on the MRR. Figure 6 shows that the MRR is significantly increased at a swinging angle of 60° and a grain size of GC #600 if a higher wire speed is used. At a higher wire speed, active grains are removed more quickly from the machining zone. However, if there are only fine GC #1000 grains in the slurry, the MRR is significantly reduced, as shown in Figure 7. If only large GC #600 grains are used, these larger grains are not properly suspended in the slurry so they deposit in the slurry and are transferred to the cutting zone

by the wire, so machining efficiency is better than that for fine GC #1000 grains. The MRR for GC #600 and GC #1000 grains increases if the working load is increased, as shown in Figure 7. The MRR for fine GC #1000 grains is between 0.5 to 1.0 mm$^3$/min because there is a smaller plowing force for specific wire-sawing conditions than for large GC #600 grains.

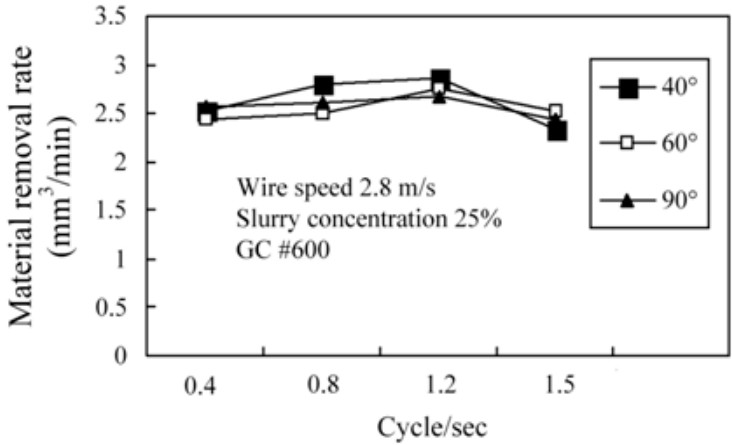

**Figure 5.** Effect of different swinging angles and frequency on MRR.

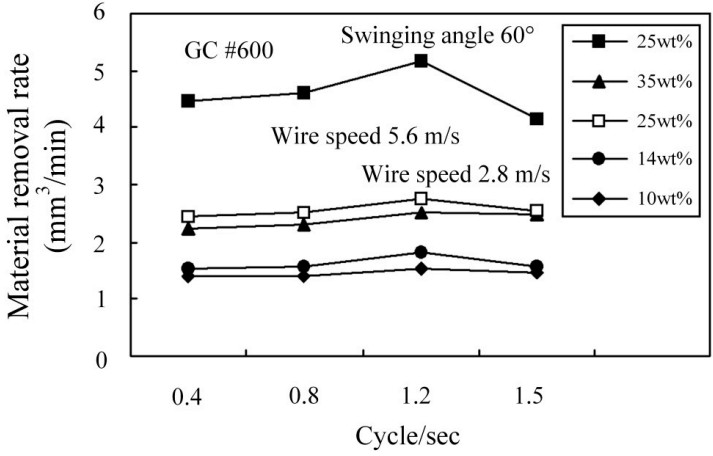

**Figure 6.** Effect of wire speed and swinging frequency on MRR.

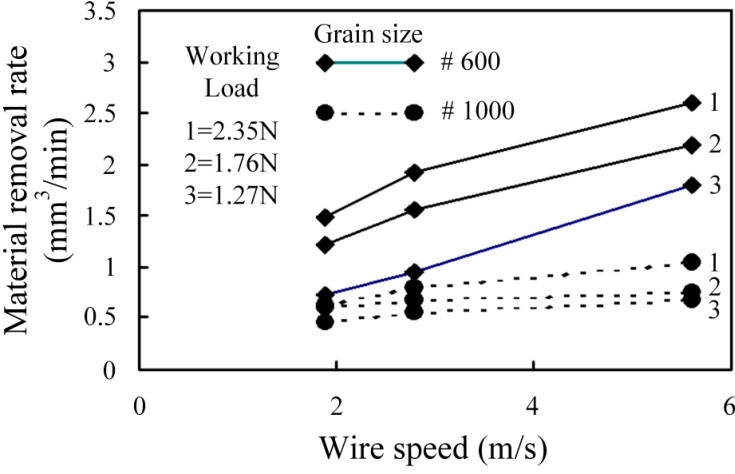

**Figure 7.** Comparison of MRR for #600 and #1000 grains.

### 3.2. *Effect of Machining Parameters on the Machined SR*

During WSM, each grain in the cutting zone exerts a cutting force that is similar to that for a polishing process, so an increase in the working load and wire speed has no effect on the machined SR. If larger grains are used, more material is removed by a single grain so the machined SR is rougher and larger cutting marks remain. Figure 8 shows that the addition of small grains into the slurry decreases the machined SR of $Al_2O_3$ and an increase in the slurry concentration has no significant effect on the machined SR of $Al_2O_3$. Figure 9 shows that there is no significant difference in the machined SR for swinging and non-swinging scenarios. Figure 10 shows that the fracture cross section for $Al_2O_3$ is rougher. Figures 11 and 12 respectively show the machined surface after WSM using a slurry of green silicon carbide (GC), with and without swinging. There are no clear cutting marks because the slurry produces an abrasive plowing force that is similar to that for a polishing process. Figure 13 shows the machined surface for wire-sawing using a fixed abrasive (diamond). Clear cutting marks are seen and the machined surface of the $Al_2O_3$ is also rougher. Diamond wire is very expensive and the diamonds on the stainless steel wire are easily lost, which renders the stainless steel wire incapable of machining, so GC cutting is not practical.

Figure 14 shows a SEM microphotograph of $Al_2O_3$ if no slurry is used. The energy-dispersive spectrometer (EDS) result for zone A (in Figure 14) is shown in Figure 15 and shows the microstructure of stainless steel, which contains Fe, Cr, Ni, and Si. If there is no slurry, the wire has no machining capability, so it becomes welded onto the workpiece surface.

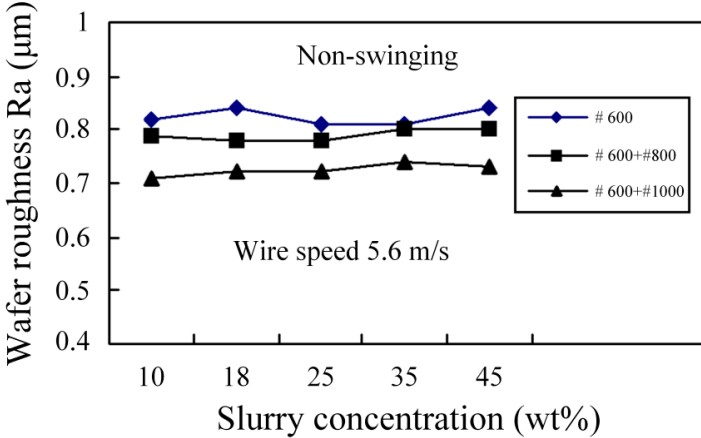

**Figure 8.** Effect of slurry concentration on machined SR for $Al_2O_3$.

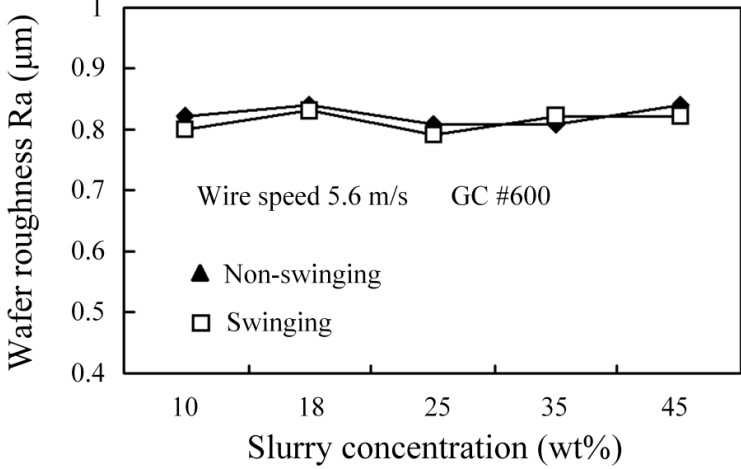

**Figure 9.** The effect swinging and non-swinging processes on the machined SR for $Al_2O_3$ for different slurry concentrations.

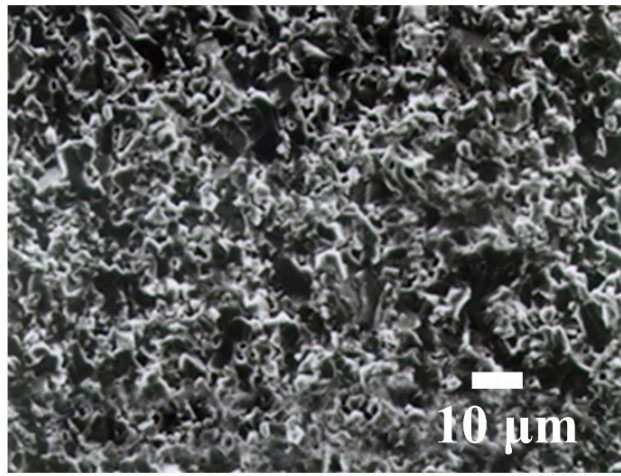

**Figure 10.** Fracture cross section for Al$_2$O$_3$.

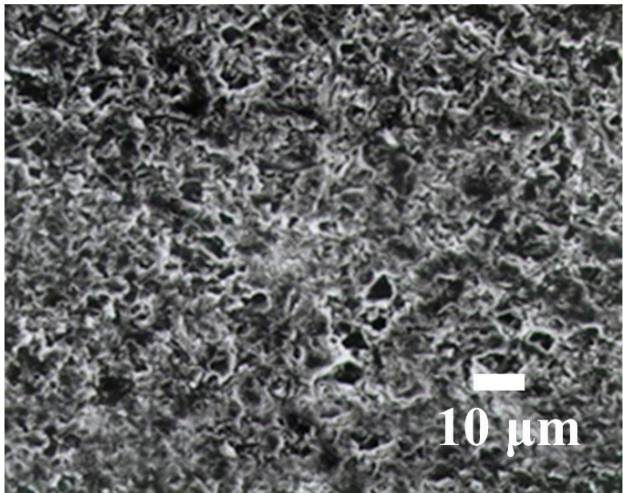

**Figure 11.** Machined surface using swinging WSM and a slurry of green silicon carbide (GC).

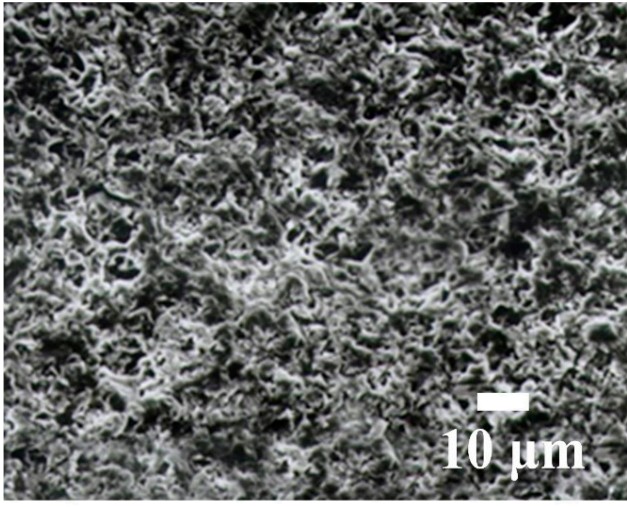

**Figure 12.** Machined surface using non-swinging WSM and a slurry (GC).

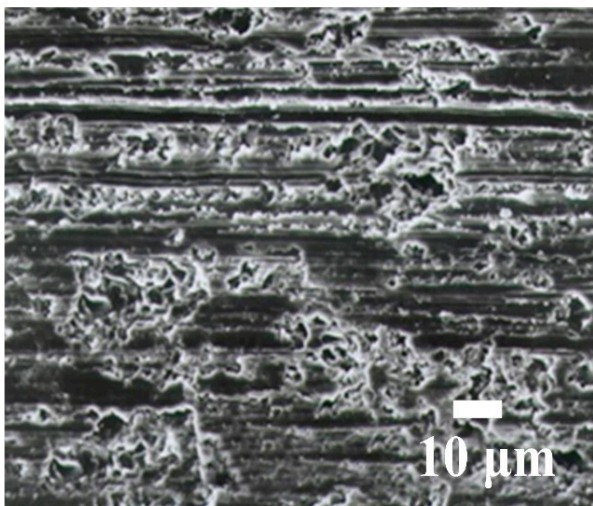

**Figure 13.** Machined surface for WSM using a fixed abrasive (diamond).

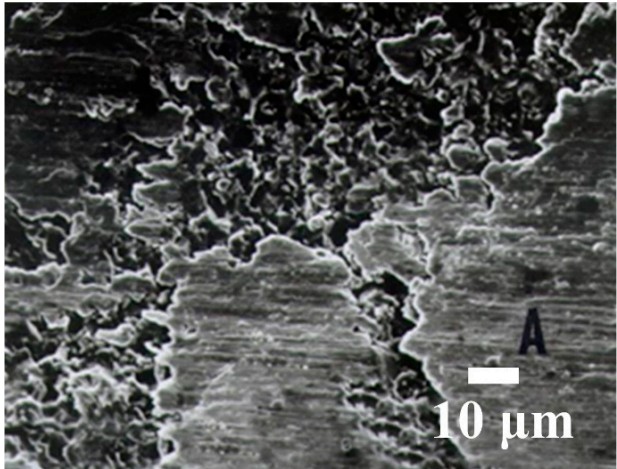

**Figure 14.** SEM photo of $Al_2O_3$ that is machined using no slurry.

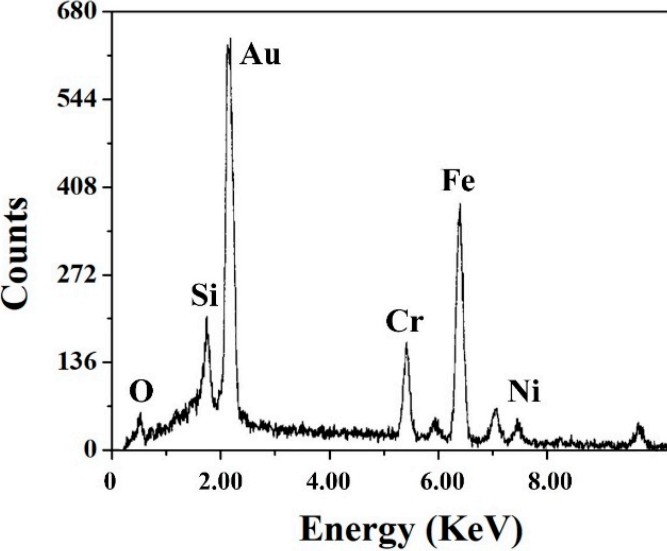

**Figure 15.** The energy-dispersive spectrometer (EDS) results for zone A.

### 3.3. Effect of Machining Parameters on Kerf Width

Figure 16 shows the experimental results for different wire speeds and concentrations. Increasing the wire speed has no significant effect on the kerf width. As the concentration increases, the number of grains that enters both sides of the stainless steel wire increases so the kerf is wider. Figure 17 shows that there is only a slight difference in the kerf width for swinging and non-swinging WSM. However, WSM with swinging produces a wider kerf than non-swinging WSM.

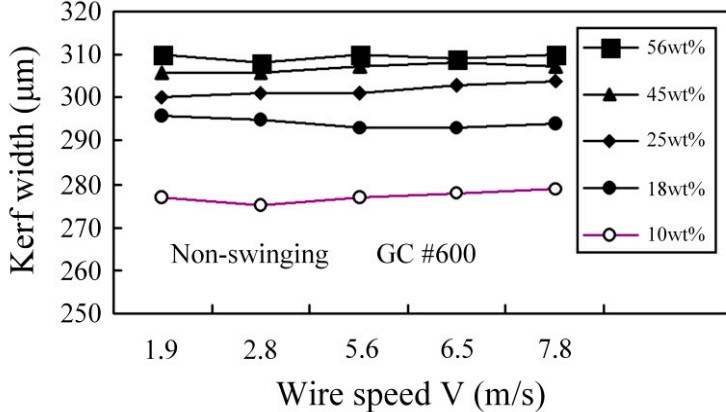

**Figure 16.** Effect of wire speed on kerf width for different slurry concentrations.

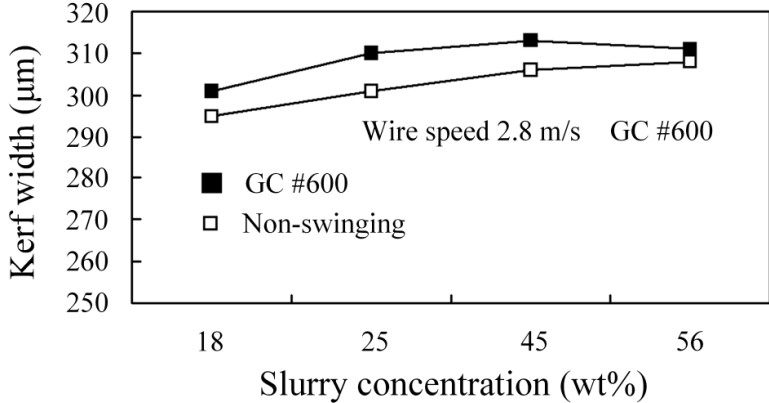

**Figure 17.** Effect of swinging and non-swinging processes on the kerf width for different slurry concentrations.

### 3.4. Effect of Machining Parameters on the Wear to Stainless Steel Wire

The degree of wire wear affects the life of the wire and the kerf width for WSM. If the working load and wire speed are increased, wire wear increases significantly because the distributed stresses (working load) that act on the stainless steel wire are more complicated and more time is required to complete WSM. If the wire speed is increased, the grains also enter and exit the cutting zone more quickly so more grains are involved in the cutting process and the wire wears more quickly, as shown in Figure 18. Wire wear is positively correlated with the MRR because the grains cut the wire as they remove material from the workpiece. A slurry concentration that gives the best MRR also produces greater wire wear. Figure 19 shows that there is no significant difference in wire wear for swinging and non-swinging processes. Figure 20 shows that for specific swinging conditions, a higher wire speed results in a significant increase in wire wear.

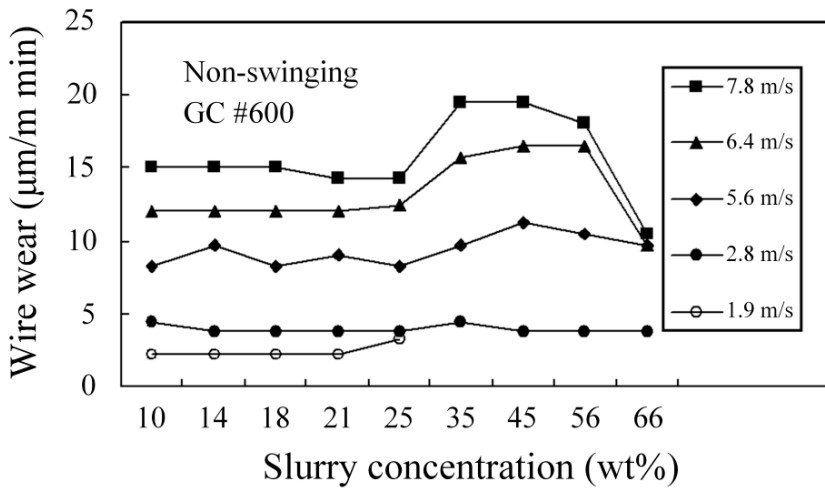

**Figure 18.** Effect of wire speed and concentration on wire wear.

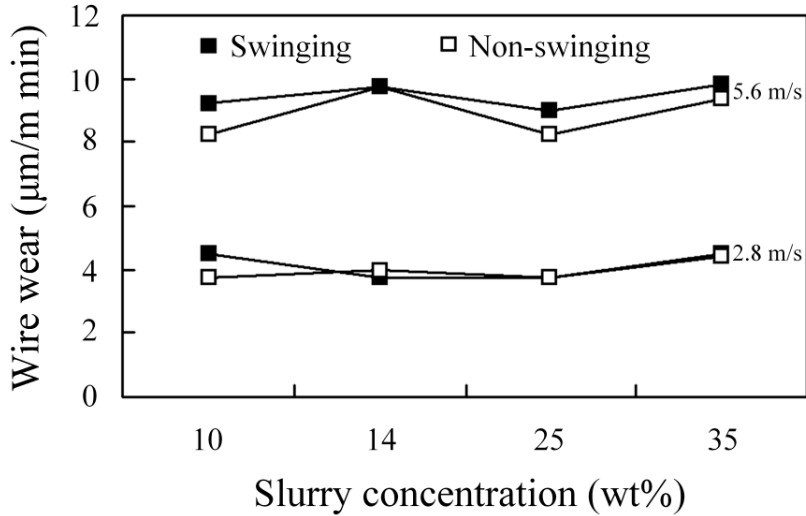

**Figure 19.** Effect of swinging and non-swinging processes on wire wear for different slurry concentrations and wire speeds.

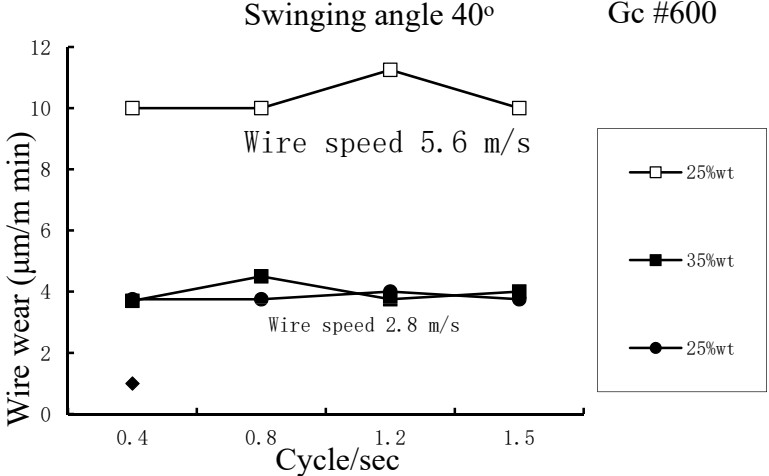

**Figure 20.** Effect of wire speed and swinging frequency on wire wear.

### 3.5. Effect of Machining Parameters on Flatness, Corner Chipping and Micro Cracking

Swinging WSM (the flatness about 2–4 μm) has no significant effect on flatness. Figure 21 shows that there is no corner chipping for WSM. Figure 22 shows SEM images of subsurface regions of $Al_2O_3$ after WSM. There is no evidence of micro cracking after WSM.

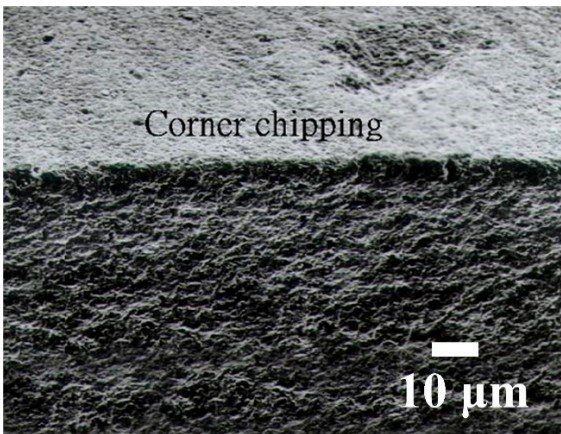

**Figure 21.** No corner chipping with swinging for WSM.

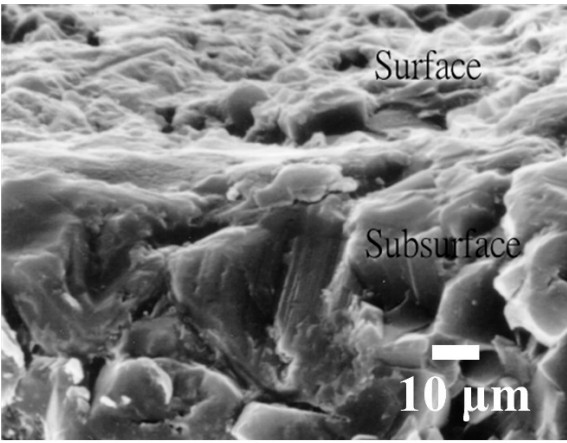

**Figure 22.** Images of subsurface regions using swinging WSM.

## 4. Conclusions

For a process that involves swinging, there is no decrease in the working load that is exerted by each active grain and less time is required for grains to enter and exit the cutting zone. The disposal of chips is also more efficient. WSM with swinging results in a higher MRR. If many active grains enter the cutting zone, grain stacking and interference prevent the grains from moving so the MRR is decreased. The cutting zone for a process that involves swinging is shorter than that for a process that does not involve swinging, so grain stacking and the peak MRR are achieved at a medium slurry concentration.

Swinging angles of 40°, 60° and 90° do not significantly affect the MRR. If only fine grains are present in the slurry, such as GC #1000 grains, the MRR is very small. If large GC #600 grains are used, these larger grains are not suspended in the slurry, so they are deposited in the slurry and are brought into the cutting zone by the wire, which gives better machining efficiency than is possible for fine GC #1000 grains.

WSM has a similar effect to a polishing process so machining conditions have no significant effect on the machined SR of $Al_2O_3$. An increase in slurry concentration results in a significant increase in kerf width and an increase in MRR. However, there is also a slight increase in wire wear for WSM.

As the grains remove parts of the workpiece, they also cut the wire. For a slurry concentration that produces a peak MRR, the wire also wears down quickly. This study shows that there is no significant difference in wire wear, flatness, corner chipping or micro cracking between processes that involve swinging and non-swinging WSM.

**Author Contributions:** Conceptualization, Y.-Y.T. and Y.-C.C.; methodology, Y.-S.L.; software, Y.-S.L.; validation, C.-C.H., Y.-Y.T. and Y.-C.C.; formal analysis, C.-C.T.; investigation, C.-C.T.; resources, C.-Y.H.; data curation, C.-Y.H.; writing—original draft and preparation, C.-Y.H.; writing—review and editing, C.-C.T.; visualization, C.-C.T.; supervision, Y.-Y.T.; project administration, Y.-Y.T.; funding acquisition, Y.-Y.T. All authors have read and agreed to the published version of the manuscript.

**Funding:** This research was funded by the Ministry of Science and Technology of the Republic of China, through Grant No. MOST 108-2622-E-262-006 -CC3, MOST 107-2221-E-002 -138 -MY2, MOST 109-2221-E-002 -087 and MOST 109-2221-E-262-003.

**Acknowledgments:** The authors would like to express their appreciation to Kinik Company for providing the slurry used in this study, and gratefully acknowledge the support of the Ministry of Science and Technology of the Republic of China.

**Conflicts of Interest:** The authors declare no conflict of interest.

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
