# Peer review of "The Effects of Different Slurry Concentrations and Wire Speeds for Swinging and Non-Swinging Wire-Saw Machining"

_processes, doi:10.3390/pr8101319_

Round 1

Reviewer 1 Report

There are many comments that refer to the size of some parameters, and there is no explanation about how they were measured or calculated.

I ask the authors to specify the following information:

  • Sample dimension;
  • For the material removal rate, what measurements were made and how the MRR was calculated for each experiment?
  • How did you calculate the length of contact between the wire and the workpiece for the swinging scenario (please explained with a sketch)?
  • How do you know the value of the working load?
  • How did you compute (measure) the wear for each experiment?
  •   For Fig. 11…. Fig 14 please specify all the parameters that characterize each cut, the magnification, the used microscope  

Author Response

Reviewer #1

  1. Sample dimension;

Response: It is revised in Table 1 according to the reviewer’s command.

Table 1. Experimental conditions.

Workpiece

Al2O3

Diameter (mm)

f8

Wire diameter (mm)

f0.240.05 (Stainless steel wire)

Slurry

Green silicon carbide (GC)+water

Grain size (mesh)

#600, #800 and #1000

Wire tension (N)

15

Concentration (wt.%)

10, 14, 25, 35, 45, 56 and 66

Wire speed (m/s)

1.9, 2.8, 5.6, 6.4 and 7.8

Working load (N)

1.27, 1.76 and 2.35

Swinging frequency (Hz)

0.4, 0.8, 1.2 and 1.5

Swinging angle (θ)

40°, 60° and 90°

  1. For the material removal rate, what measurements were made and how the MRR was calculated for each experiment?

Response: It is revised in Eq. (1) for MRR according to the reviewer’s command.

The MRR for each SWSM can be calculated using Eq. (1), which are defined as follows:

                                                                                                          (1)

where is the original weight before SWSM and is the actual weight after SWSM, and t is the each period of SWSM.

  1. How did you calculate the length of contact between the wire and the workpiece for the swinging scenario (please explained with a sketch)?

Response: The length of contact between the wire and the workpiece for the swinging is not a constant due to workpiece is always swinging. Fig. 2 (a) shows the workpiece (J) also swings back and forth, Fig. 2 (b) shows the length of contact between the wire and the workpiece during WSM.

(a)

(b)

Figure 2. (a) Model of swinging wire-sawing (b) contact between the wire and the workpiece.

  1. How do you know the value of the working load?

Response: A tension meter (IMADA, DPRSX-10TR) was used to measure the wire tension and working load. It is revised according to the reviewer’s command.

  1. How did you compute (measure) the wear for each experiment?

Response: It is revised in Eq. (2) for wire wear (WW) according to the reviewer’s command.

It is revised in Eq. (2) for wire wear (WW) according to the reviewer’s command.

The wire wear (WW) for each SWSM can be calculated using Eq. (2), which are defined as follows:

                                                                                                           (2)

where is the original diameter before SWSM, is the diameter after SWSM and t is the each period of SWSM.

  1. For Fig. 11…. Fig 14 please specify all the parameters that characterize each cut, the magnification, the used microscope.

Response: It is added the magnification for Figs.10-14 according to the reviewer’s command.

Figure 10. Fracture cross section of Al2O3.

Figure 11. Machined surface using swinging WSM and a slurry (GC).

Figure 12. Machined surface using non-swinging WSM and a slurry (GC).

Figure 13. Machined surface for WSM using a fixed abrasive (diamond).

Figure 14. SEM photo of Al2O3 that is machined using no slurry.

Figure 15. The energy-dispersive spectrometer (EDS) for zone A.

Reviewer 2 Report

The manuscript seems well but few improvements or comments are necessary.

1) Page 1, line 36 - the predicate is to be "produce", not "produced"

2) Page 2, line 45 - the word "made" or another part of predicate is missing.

3) Page 2, line 54 - the predicate needs to be "is subjected".

4) Page 2, lines 74-76 - the sentence beginning "Xu et al. [13] found..." need to be improved - the meaning of the central part of the sentence is unclear. Maybe, it should be better to divide it into two parts, one for "unimportantly" and the second for "importantly".

5) Figure 3 - Why are the values for slurry concentrations 45 and 56 wt% and wire speed 1.9 m/s missing? Add missing values or explain this absence, please.

6) Figure 4 - Why are the values for slurry concentrations 45 and 56 wt% and swinging processes missing? Add missing values or explain this absence, please.

7) Page 6, line 145 - maybe, it could be better to use instead of "a larger cutting mark remains" the formulation "larger cutting marks remain".

8) Page 6, lines 146 and 147 - the use of the indefinite article for Al2O3 in the second part of the sentence after using the definite article in the first part of the sentence is inappropriate.

9) Page 6, lines 153-155 - the end of the sentence is unclear regarding the previous statements - GC is used for cutting with the slurry of "green silicon carbide" and due to my meaning, authors can document benefits of this technology process comparing to diamond wire sawing; therefore, the note that "GC cutting is not practical" seems to be incorrect.

10) Please, explain the highest peak in the Figure 15 indicated as "Aurum". Where is the source of this element in the process?

11) Figure 18 - Why are the values for slurry concentrations 35, 45, 56 and 66 wt% and wire speed 1.9 m/s missing? Add missing values or explain this absence, please.

12) Figure 20 - the attempt to put a lot of information into one graph resulted to unclear presentation. The description inside the graph is quite unclear, because it is not evident to which data is related (Vibration angle 40° and GC #600) - moreover, values for 14wt% are not visible, presentation of "25wt%" in the bottom of the graph is completely unclear. Why are not presented more wt% for higher wire speed? 

13) Page 12, line 228 - it should be used "better" instead of "good".

Author Response

Reviewer #2

  1. Page 1, line 36 - the predicate is to be "produce", not "produced"

Response: It is revised according to the reviewer’s command.

2) Page 2, line 45 - the word "made" or another part of predicate is missing.

Response: It is revised according to the reviewer’s command.

3) Page 2, line 54 - the predicate needs to be "is subjected".

Response: It is revised according to the reviewer’s command.

4) Page 2, lines 74-76 - the sentence beginning "Xu et al. [13] found..." need to be improved - the meaning of the central part of the sentence is unclear. Maybe, it should be better to divide it into two parts, one for "unimportantly" and the second for "importantly".

Response: It is revised according to the reviewer’s command.

5) Figure 3 - Why are the values for slurry concentrations 45 and 56 wt% and wire speed 1.9 m/s missing? Add missing values or explain this absence, please.

Response: The MRR decreases at wire speed of 1.9 m/s because the wire speed of 1.9 m/s is not allowed active grains to enter and exit at 45 and 50 wt% slurry concentration.

6) Figure 4 - Why are the values for slurry concentrations 45 and 56 wt% and swinging processes missing? Add missing values or explain this absence, please.

Response: The MRR decreases at swinging processes because the active grains are not allowed to enter and exit the cutting zone.

7) Page 6, line 145 - maybe, it could be better to use instead of "a larger cutting mark remains" the formulation "larger cutting marks remain".

Response: It is revised according to the reviewer’s command.

8) Page 6, lines 146 and 147 - the use of the indefinite article for Al2O3 in the second part of the sentence after using the definite article in the first part of the sentence is inappropriate.

Response: It is revised according to the reviewer’s command.

9) Page 6, lines 153-155 - the end of the sentence is unclear regarding the previous statements - GC is used for cutting with the slurry of "green silicon carbide" and due to my meaning, authors can document benefits of this technology process comparing to diamond wire sawing; therefore, the note that "GC cutting is not practical" seems to be incorrect.

Response: It is revised according to the reviewer’s command.

10) Please, explain the highest peak in the Figure 15 indicated as "Aurum". Where is the source of this element in the process?

Response: It is a misunderstanding. In order to understand the microstructure of wire saw (stainless steel), a layer of gold (Au) film needs to be coated on the wire saw. So, the component of gold appears in Fig. 15.

11) Figure 18 - Why are the values for slurry concentrations 35, 45, 56 and 66 wt% and wire speed 1.9 m/s missing? Add missing values or explain this absence, please.

Response: See the response of problem 5.

12) Figure 20 - the attempt to put a lot of information into one graph resulted to unclear presentation. The description inside the graph is quite unclear, because it is not evident to which data is related (Vibration angle 40° and GC #600) - moreover, values for 14wt% are not visible, presentation of "25wt%" in the bottom of the graph is completely unclear. Why are not presented more wt% for higher wire speed?

Response: It is deleted 10wt% and 14wt% (Vibration angle 40° and GC #600) from Figure 20 according to the reviewer’s command.

13) Page 12, line 228 - it should be used "better" instead of "good".

Response: It is revised according to the reviewer’s command.
